



**Evaluating Nitrogen Cycling in Terrestrial Biosphere Models: Implications for the Future**
**Terrestrial Carbon Sink**
Sian Kou-Giesbrecht[1], Vivek Arora[1], Christian Seiler[2], Almut Arneth[3], Stefanie Falk[4], Atul Jain[5],
Fortunat Joos[6], Daniel Kennedy[7], Jürgen Knauer[8], Stephen Sitch[9], Michael O'Sullivan[9], Naiqing
Pan[10], Qing Sun[6], Hanqin Tian[10], Nicolas Vuichard[11], and Sönke Zaehle[12]
[1]Canadian Centre for Climate Modelling and Analysis, Climate Research Division, Environment
Canada, Victoria, Canada
[2]School of Environmental Studies, Queen's University, Kingston, Canada
[3]Karlsruhe Institute of Technology, Atmospheric Environmental Research, Garmisch-
Partenkirchen, Germany
[4]Department für Geographie, Ludwig-Maximilians-Universität Munich, München, Germany
[5]Department of Atmospheric Sciences, University of Illinois Urbana-Champaign, Urbana, USA
[6]Climate and Environmental Physics, Physics Institute and Oeschger Centre for Climate Change
Research, University of Bern, Bern, Switzerland
[7]National Center for Atmospheric Research, Climate and Global Dynamics, Terrestrial Sciences
Section, Boulder, USA
[8]Hawkesbury Institute for the Environment, Western Sydney University, Penrith, Australia
[9]Faculty of Environment, Science and Economy, University of Exeter, Exeter, UK
[10]Schiller Institute for Integrated Science and Society, Department of Earth and Environmental
Sciences, Boston College, Chestnut Hill, USA
[11]Laboratoire des Sciences du Climat et de l'Environnement, LSCE-IPSL (CEA-CNRS-UVSQ),
Université Paris-Saclay, Gif-sur-Yvette, France
[12]Max Planck Institute for Biogeochemistry, Jena, Germany
*Correspondence to:* Sian Kou-Giesbrecht (sian.kougiesbrecht@ec.gc.ca)
**Abstract**
Terrestrial carbon (C) sequestration is limited by nitrogen (N), a constraint that could
intensify under $CO_2$ fertilisation and future global change. The terrestrial C sink is estimated to
currently sequester approximately a third of annual anthropogenic $CO_2$ emissions based on an
ensemble of terrestrial biosphere models, which have been evaluated in their ability to reproduce
observations of the C, water, and energy cycles. However, their ability to reproduce observations
of N cycling and thus the regulation of terrestrial C sequestration by N has been largely
unexplored. Here, we evaluate an ensemble of terrestrial biosphere models with coupled C-N
cycling and their performance at simulating N cycling, outlining a framework for evaluating N
cycling that can be applied across terrestrial biosphere models. We find that models exhibit





significant variability across N pools and fluxes, simulating different magnitudes and trends over
the historical period, despite their ability to generally reproduce the historical terrestrial C sink.
This suggests that the underlying N processes that regulate terrestrial C sequestration operate
differently across models and may not be fully captured. Furthermore, models tended to
overestimate tropical biological N fixation, vegetation C:N ratio, and soil C:N ratio but
underestimate temperate biological N fixation relative to observations. However, there is
significant uncertainty associated with measurements of N cycling processes given their scarcity
(especially relative to those of C cycling processes) and their high spatiotemporal variability.
Overall, our results suggest that terrestrial biosphere models that represent coupled C-N cycling
(let alone those without a representation of N cycling) could be overestimating C storage per unit
N, which could lead to biases in projections of the future terrestrial C sink under $CO_2$ fertilisation
and future global change. More extensive observations of N cycling processes are crucial to
evaluate N cycling and its impact on C cycling as well as guide its development in terrestrial
biosphere models.
**Plain Language Summary**

Nitrogen (N) is an essential limiting nutrient to terrestrial carbon (C) sequestration. We
evaluate N cycling in an ensemble of terrestrial biosphere models. We find that they simulate
significant variability in N processes. Models tended to overestimate C storage per unit N in
vegetation and soil, which could have consequences for projecting the future terrestrial C sink.
However, N cycling measurements are highly uncertain and more are necessary to guide the
development of N cycling in models.

**1 Introduction**

The terrestrial biosphere is estimated to currently sequester approximately a third of
anthropogenic $CO_2$ emissions by the Global Carbon Project (GCP) (Friedlingstein et al., 2022).
The GCP annually reports an estimate of the global carbon (C) budget which includes an
estimate of the atmosphere-land $CO_2$ flux, i.e., the terrestrial C sink, based on simulations of an
ensemble of terrestrial biosphere models – the trends in the land carbon cycle project (TRENDY)
ensemble. In recent years, the majority of the models within the TRENDY ensemble have
incorporated a representation of coupled C and nitrogen (N) cycling given the empirically
established importance of N limitation of vegetation growth (Elser et al., 2007; Fernández-
Martínez et al., 2014; LeBauer and Treseder, 2008; Wright et al., 2018): whereas only four out of
nine models represented coupled C-N cycling in the 2013 GCP, 11 out of 16 models represented
coupled C-N cycling in the 2022 GCP (Figure 1). Capturing N constraints on C cycling is critical
for realistically simulating the terrestrial C sink, which arises from the combined effects of
concurrently acting global change drivers that are each modulated by N: $CO_2$ fertilisation is
limited by N (Terrer et al., 2019; Wang et al., 2020a), intensifying N deposition increases N
supply (O'Sullivan et al., 2019; Wang et al., 2017), rising temperature and varying precipitation
modulate decomposition and soil N availability (Liu et al., 2017), and land use change and
associated N fertilisation regimes determine N supply to crops.





Figure 1: Inclusion of coupled C-N cycling in the terrestrial biosphere models contributing to the
Global Carbon Project, i.e., the TRENDY ensemble.

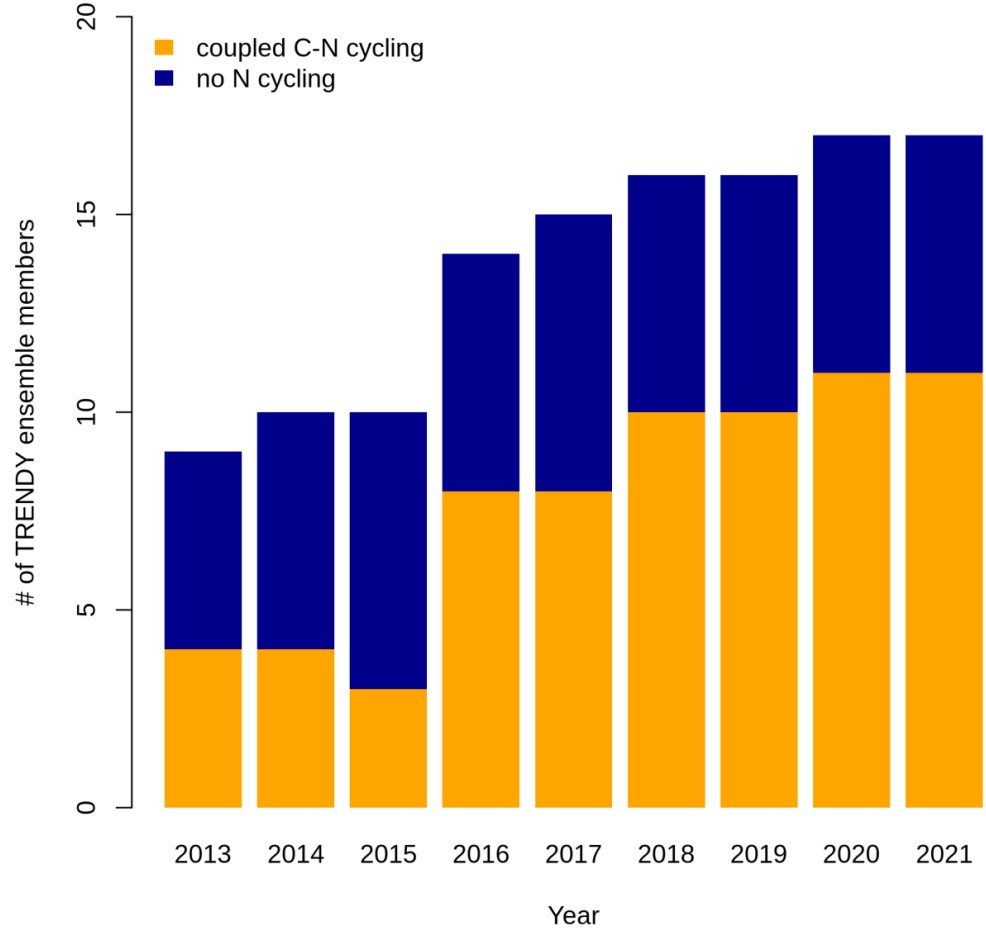




The TRENDY ensemble has been extensively evaluated against observations of the C,
water, and energy cycles (Collier et al., 2018; Friedlingstein et al., 2022; Seiler et al., 2022).
Within the GCP itself, the primary simulated C pools, C fluxes, and water fluxes are evaluated
using a skill score system developed by the International Land Model Benchmarking Project
(ILAMB) that quantifies model performance by comparing model simulations to observations
(Collier et al., 2018; Friedlingstein et al., 2022). ILAMB scores encompass the mean and
variability of a given pool or flux over monthly to decadal temporal scales and over grid cell to
global spatial scales. However, N cycling has not been explicitly evaluated despite its importance
in regulating C cycling. This is in part due to the relatively recent incorporation of N cycling in
terrestrial biosphere models (Figure 1) (Fisher and Koven, 2020; Hungate et al., 2003) but also
due to the paucity of global observation-based datasets of N cycling: N exists in many forms and
is lost from terrestrial ecosystems via numerous pathways (emissions of $NH_3$, $N_2O$, $NO_x$ and $N_2$
as well as $NO_3^-$ and $NH_4^+$ leaching), N processes are generally not measured in situ in networks
such as FLUXNET, and remote sensing methodologies for measuring N processes are still in
their infancy. Additionally, N processes exhibit extremely high spatial and temporal variabilities
and are thus challenging to measure. As such, N cycling has commonly been evaluated by
comparing simulated N pools and fluxes to global totals based on a small number of observations
that have been scaled up or averaged to yield a value with wide confidence intervals (Davies-
Barnard et al., 2020).
N cycling is implicitly evaluated by comparing terrestrial biosphere models without N
cycling to those with coupled C-N cycling in reproducing observations of the C, water, and
energy cycles in the absence of N cycle observations. Results suggest that there are only minor
differences between the performance of models with and without N cycling. There is no
significant difference between the terrestrial C sink simulated by the TRENDY models with and
without N cycling (Friedlingstein et al., 2022) nor between the terrestrial C sink simulated by the
models participating in the Multi-scale synthesis and Terrestrial Model Intercomparison Project
(MsTMIP) with and without N cycling (Huntzinger et al., 2017). Comparing the mean score
across all C, water, and energy cycle variables between TRENDY models with and without N
cycling yielded no significant difference (Seiler et al., 2022). However, TRENDY models
without N cycling had significantly higher scores for net biome productivity than TRENDY
models with N cycling (although all other variables were not significantly different between
TRENDY models with and without N cycling, including vegetation C, soil C, net biome
productivity, leaf area index, latent heat flux, and runoff, among others) (Seiler et al., 2022).
Despite this seeming absence of a difference between models with and without coupled C-N
cycling in simulating the current terrestrial C sink, it is imperative that N constraints on C
cycling are properly represented by terrestrial biosphere models in order to realistically simulate
the terrestrial C sink under future global change, which modifies the C/N balance through N
limitation of $CO_2$ fertilisation and intensifying N deposition among other effects of global
change. As such, explicitly evaluating N cycling processes themselves is necessary to assess the
ability of terrestrial biosphere models to capture the underlying mechanisms that determine
terrestrial C sequestration and thus to realistically project the future terrestrial C sink under
global change.



Here, we synthesise the N pools and fluxes simulated by 11 terrestrial biosphere models
in the TRENDY ensemble that participated in the 2022 GCP. We evaluate their performance in
reproducing observations of three key variables of the N cycle: biological N fixation, vegetation
C:N ratio, and soil C:N ratio. These three variables are critical to C cycling because (1)
biological N fixation is the dominant natural N supply to terrestrial ecosystems, influencing the
degree of N limitation of plant growth and thus terrestrial C sequestration, and (2) vegetation and
soil C:N ratios reflect assimilated C per unit N and thus terrestrial C storage.

**2 Methods**
**2.1 Simulation Protocol**
For the 2022 GCP (version 11), the TRENDY ensemble consisted of 16 terrestrial
biosphere models, 11 of which represent N cycling (CABLE-POP, CLM5.0, DLEM, ISAM,
JSBACH, JULES-ES, LPJ-GUESS, LPX-Bern, OCNv2, ORCHIDEEv3, and SDGVM).
Although SDGVM includes a representation of N cycling, its representation is simplistic and was
therefore not included. Additionally, CLASSIC contributed to the 2022 GCP without coupled C-
N cycling; the S3 simulation was repeated by CLASSIC with coupled C-N cycling following the
2022 GCP protocol and was used here. Overall, we analysed eleven models with coupled C-N
cycling (Table 1).



Table 1: Terrestrial biosphere models in the TRENDY-N ensemble and descriptions of their
representations of N limitation of vegetation growth, biological N fixation, vegetation response
to N limitation (i.e., strategies in which vegetation invests C to increase N supply in N-limited
conditions), and N limitation of decomposition.

| | Reference | N limitation of vegetation growth | Biological N fixation | Vegetation response to N limitation | N limitation of decomposition |
|---|---|---|---|---|---|
| CABLE-POP | (Haverd et al., 2018) | $V_{cmax}$ flexible C:N stoichiometry | Time-invariant | Static | N-invariant |
| CLASSIC | (Melton et al., 2020) | $V_{cmax}$ flexible C:N stoichiometry | f(N limitation of vegetation growth) | Dynamic (biological N fixation) | N-invariant |
| CLM5.0 | (Lawrence et al., 2019) | $V_{cmax}$ flexible C:N stoichiometry | f(N limitation of vegetation growth) | Dynamic (biological N fixation, mycorrhizae, retranslocation) | f(soil N) |
| DLEM | (Tian et al., 2015) | GPP | f(soil T, soil $H_2O$, soil C, soil N) | Dynamic (root allocation) | f(soil N) |
| ISAM | (Shu et al., 2020) | GPP | f(ET) | Static | f(soil N) |
| JSBACH | (Reick et al., 2021) | NPP | f(NPP) | Static | f(soil N) |
| JULES-ES | (Wiltshire et al., 2021) | NPP | f(NPP) | Static | f(soil N) |
| LPJ-GUESS | (Smith et al., 2014) | $V_{cmax}$ flexible C:N stoichiometry | f(ET) | Dynamic (root allocation) | N-invariant |
| LPX-Bern | (Lienert and Joos, 2018) | NPP | Derived post hoc to simulate a closed N cycle | Static | N-invariant |
| OCNv2 | (Zaehle and Friend, 2010) | $V_{cmax}$ flexible C:N stoichiometry | f(N limitation of vegetation growth) | Dynamic (root allocation) | f(soil N) |



| ORCHIDEEv3 | (Vuichard et al., 2019) | $V_{cmax}$ flexible C:N stoichiometry | Time-invariant | Static | N-invariant |
|---|---|---|---|---|---|




We analysed the S3 simulation which includes historical changes in atmospheric $CO_2$,
climate, N deposition, N fertilisation, and land use from 1851 to 2021 (see Friedlingstein et al.
(2022) for a full description of the simulation protocol). Briefly, models were forced with
atmospheric $CO_2$ from Dlugokencky and Tans (2022), the merged monthly Climate Research
Unit (CRU) and 6-hourly Japanese 55-year Reanalysis (JRA-55) dataset or the monthly CRU
dataset Harris et al. (2020), N deposition from Hegglin et al. (2016) / Tian et al. (2022), N
fertilisation from the global $N_2O$ Model Intercomparison Project (NMIP) (Tian et al., 2018), and
land use from the LUH2-GCB2022 (Land-Use Harmonization 2) dataset (Chini et al., 2021;
Hurtt et al., 2020; Klein Goldewijk et al., 2017a, b). We interpolated outputs from all models to a
common resolution of 1° x 1° using bilinear interpolation.

**2.2 Terrestrial biosphere model descriptions**

The terrestrial biosphere models in the TRENDY ensemble employ a wide variety of
assumptions and formulations of N cycling processes, reflecting knowledge gaps and divergent
theories (Table 1). Here we describe four fundamental aspects of N cycling for each terrestrial
biosphere model: N limitation of vegetation growth, biological N fixation, the response of
vegetation to N limitation (i.e., strategies in which vegetation invests C to increase N supply in
N-limited conditions), and N limitation of decomposition. These have been identified as
important challenges for representing N cycling in terrestrial biosphere models (Meyerholt et al.,
2020; Peng et al., 2020; Stocker et al., 2016; Wieder et al., 2015a; Zaehle et al., 2015; Zaehle and
Dalmonech, 2011).
Terrestrial biosphere models differ in how N limitation of vegetation growth is
represented (Thomas et al., 2015). Some TRENDY models represent flexible C:N stoichiometry
and modelled maximum carboxylation rate of photosynthesis ($V_{cmax}$) decreases with decreasing
leaf N (CABLE-POP, CLASSIC, CLM5.0, LPJ-GUESS, OCNv2, ORCHIDEEv3) following
empirical evidence (Walker et al., 2014). Other TRENDY models represent time-invariant C:N
stoichiometry and modelled GPP or NPP decreases with N limitation (DLEM, ISAM, JSBACH,
JULES-ES, and LPX-Bern). Importantly, flexible vs. time-invariant C:N stoichiometry
determines terrestrial C storage per unit N.
Biological N fixation is the dominant natural N supply to terrestrial ecosystems (Vitousek
et al., 2013). In terrestrial biosphere models, biological N fixation has generally been represented
phenomenologically as a function of either net primary productivity (NPP) or evapotranspiration
(ET) (Cleveland et al., 1999). More recently, representations of biological N fixation have been
updated such that it is up-regulated in N-limited conditions following empirical evidence (Menge
et al., 2015; Vitousek et al., 2013; Zheng et al., 2019). The majority of TRENDY models
represent biological N fixation phenomenologically (ISAM, JSBACH, JULES-ES, and LPJ-
GUESS). Three TRENDY models (CLASSIC, CLM5.0, and OCNv2) represent biological N
fixation mechanistically such that it increases with N limitation of vegetation (Kou-Giesbrecht
and Arora, 2022; Lawrence et al., 2019; Meyerholt et al., 2016). These representations separate
free-living biological N fixation (via soil microbes, epiphytic microbes, lichens, bryophytes, etc.
(Reed et al., 2011)) from symbiotic biological N fixation, which is regulated by N limitation of
vegetation. DLEM derives biological N fixation as a function of soil temperature, soil moisture,





soil C, and soil N. LPX-Bern derives biological N fixation post hoc to simulate a closed N cycle,
implicitly including rock N sources (Joos et al., 2020). Finally, CABLE-POP and ORCHIDEEv3
represent biological N fixation as a specified time-invariant input over the historical period.
Importantly, representing the regulation of biological N fixation by N limitation does not only
determine biological N fixation itself but also modulates terrestrial C sequestration: it enables
vegetation to increase N uptake in N-limited conditions, reduce N limitation, and thus sustain
terrestrial C sequestration. Some TRENDY models (DLEM, LPJ-GUESS, and OCNv2) also
represent increasing C allocation to roots with increasing N limitation (Smith et al., 2014; Zaehle
and Friend, 2010) following empirical evidence (Poorter et al., 2012). This enables vegetation to
increase root N uptake in N-limited conditions, reduce N limitation, and thus sustain terrestrial C
sequestration. The response of vegetation to N limitation, which could also include increased C
allocation to mycorrhizae (Phillips et al., 2013) (represented in CLM5.0) or increased
retranslocation of N during tissue turnover (Du et al., 2020; Han et al., 2013; Kobe et al., 2005)
(represented in CLM5.0) is important for determining terrestrial C sequestration.
Decomposition rate is controlled by soil temperature, soil moisture, and N content in
litter, where increasing litter C:N ratio decreases decomposition rate (Cotrufo et al., 2013). Some
TRENDY models represent this reduction in decomposition rate with increasing litter C:N ratio
(CLM5.0, DLEM, ISAM, JSBACH, JULES-ES, and OCNv2) following empirical evidence.

**2.3 Observation-based datasets**

We interpolated observation-based datasets to a common resolution of 1° x 1° using
bilinear interpolation for comparison against model outputs. To compare model outputs against
observation-based datasets we averaged model outputs over 1980–2021, which spans the period
in which most measurements were made.

**2.3.1 Biological N fixation**

A biological N fixation observation-based dataset was derived from Davies-Barnard and
Friedlingstein (2020), a global meta-analysis of field measurements of natural biological N
fixation (free-living and symbiotic) that scales biome-specific means onto the Collection 5
MODIS Global Land Cover Type International Geosphere-Biosphere Programme (IGBP)
product (Friedl et al., 2010). To account for agricultural biological N fixation, we assumed that
N-fixing crops account for 15.7% of global cropland area (U.S. Department of Agriculture,
2022) and their biological N fixation rate as 11.5 g N m$^{-2}$ yr$^{-1}$ (Herridge et al., 2008). We
assumed that N-fixing crops are distributed evenly across all cropland. We amended the dataset
from Davies-Barnard and Friedlingstein (2020) to include agricultural biological N fixation
(DBF-USDA).
The score of LPX-Bern in simulating biological N fixation is not analysed because it
implicitly includes rock N sources and is thus not directly comparable to the observation-based
dataset.

**2.3.2 Vegetation C:N ratio**



A vegetation C:N ratio observation-based dataset was derived by scaling biome-specific
means from the TRY plant trait database (Kattge et al., 2020) onto the Collection 5 MODIS
Global Land Cover Type IGBP product (Friedl et al., 2010). First, we obtained N content per dry
mass for leaves, root, and stem, as well as C content per dry mass for leaves, root, and stem from
the TRY plant trait database. We selected entries that reported species. Second, we obtained
plant functional type (PFT) for each species from the TRY plant trait database. We categorised
each PFT into the IGBP land cover types (Table A1) and then used this to categorise each entry
into the IGBP land cover types. We averaged across entries in each IGBP land cover type. Third,
we divided mean tissue C content per tissue dry mass by mean tissue N content per tissue dry
mass for each tissue and for each IGBP land cover type. Fourth, we weighed each tissue by its
PFT-specific fraction of total biomass from Poorter et al. (2012) to obtain total vegetation C:N
ratio for each IGBP land cover type. Lastly, we scaled total vegetation C:N ratio for each IGBP
land cover type to the Collection 5 MODIS Global Land Cover Type IGBP product.
**2.3.3 Soil C:N ratio**
A soil C:N ratio observation-based dataset was derived from soil C and soil N products
from SoilGrids (Poggio et al., 2021), which provides globally gridded datasets of soil organic C
and total soil N at a 250m x 250m resolution for six layers up to a depth of 200 cm. These
estimates are derived using machine learning methods and soil observations from 240 000
locations across the globe and over 400 environmental covariates. We summed soil C over all
layers and soil N over all layers (using the bulk density and depth of each layer) then obtained
the soil C:N ratio.
**2.3.4 C cycling variables**
In addition to evaluating N cycling variables, we also evaluated the primary C cycling
variables: gross primary productivity (GPP), net biome productivity (NBP), vegetation C
(CVEG), soil C (CSOIL), and leaf area index (LAI). These variables have been previously
evaluated in detail for the terrestrial biosphere models in the TRENDY ensemble (GCP 2021) in
Seiler et al. (2022). Seiler et al. (2022) gives further details on the observation-based datasets
used to evaluate the primary C cycling variables. Briefly, we evaluated GPP against MODIS
(Zhang et al., 2017), GOSIF (Li and Xiao, 2019), and FLUXCOM (Jung et al., 2020) products.
We evaluated NBP against the CAMS (Agustí-Panareda et al., 2019), CarboScope (Rödenbeck
et al., 2018), and CT2019 (Jacobson et al., 2020) products. We evaluated CVEG against the
GEOCARBON (Avitabile et al., 2016; Santoro et al., 2015), Zhang and Liang (2020), and Huang
et al. (2021) products. We evaluated LAI against AVHRR (Claverie et al., 2016), Copernicus
(Verger et al., 2014), and MODIS (Myneni et al., 2002) products. We evaluated CSOIL against
HWSD (Todd-Brown et al., 2013; Wieder, 2014) and SoilGrids (Hengl et al., 2017) products.
These observation-based products are globally gridded.
**2.4 Model evaluation with the Automated Model Benchmarking R Package (AMBER)**
The Automated Model Benchmarking R (AMBER) package developed by Seiler et al.
(2021) quantifies model performance in reproducing observation-based datasets using a skill
score system that is based on ILAMB (Collier et al., 2018). Five scores assess the simulated





time-mean bias ($S_{bias}$), monthly centralised root-mean-square-error ($S_{rmse}$), seasonality ($S_{phase}$),
inter-annual variability ($S_{iav}$), and spatial distribution ($S_{dist}$) in comparison to the observation-
based dataset. Scores are dimensionless and range from 0 to 1, where higher values indicate
better model performance. The overall score for each variable ($S_{overall}$) is
$$S_{overall} = \text{mean}\left(S_{bias}, S_{rmse}, S_{phase}, S_{iav}, S_{dist}\right)$$

We calculated the overall score for each C and N cycling variable. Because biological N fixation,
vegetation C:N ratio, and soil C:N ratio datasets are representative of the present-day (as a single
time point), $S_{rmse}$, $S_{phase}$, and $S_{iav}$ are not defined and thus do not contribute to $S_{overall}$. This also
holds for vegetation C and soil C. The calculation of each score is described in detail in Seiler et
al. (2022).

**2.5 Statistics**

We used a Mann-Kendall trend test to assess the existence of a statistically significant
trend in the time series over the historical period for simulated C and N cycling variables (Hipel
and McLeod, 1994). We calculated Spearman's rank correlation coefficient to assess the
existence of statistically significant correlations between overall scores, present-day global
values, and Kendall's tau. We used a t-test or ANOVA (p-value < 0.05) to assess the existence of
statistically significant differences between overall scores, present-day global values, and
Kendall's tau for models with different representations of N limitation of vegetation growth,
biological N fixation, vegetation response to N limitation, and N limitation of decomposition
(Table 1).

**3 Results**

**3.1 Net biome productivity**

Figure 2 shows NBP simulated by the TRENDY ensemble models with coupled C-N
cycling (hereafter referred to as the TRENDY-N ensemble). NBP is the difference between the
net natural atmosphere-land flux of $CO_2$ and land use change $CO_2$ emissions. Positive values of
NBP indicate a terrestrial C sink whereas negative values of NBP indicate a terrestrial C source.
All TRENDY-N ensemble models suggest a terrestrial C sink for the present-day, agreeing with
the global C budget constraint from the 2022 Global C Budget with most models within two
standard deviations of the mean (1.5 ± 0.6 Pg C for 2012–2021) (Figure 2a). The TRENDY-N
ensemble agrees reasonably well with observations globally, agreeing somewhat better with
CarboScope and CT2019 than with CAMS (Figure 2b). However, the latitudinal distributions of
the observation-based datasets display weak agreement among themselves, with opposing signs
in multiple regions, especially at southern latitudes and at high northern latitudes (Figure 2b).
This is in part due to the smaller land area at these latitudes. The region showing the strongest
agreement is mid to high northern latitudes, in which both the TRENDY-N ensemble and
observations suggest a terrestrial C sink (Figure 2b).



Figure 2: Net biome productivity (NBP) simulated by the TRENDY-N ensemble. a. Global NBP
from 1960 to 2021. The boxes indicate the global C budget constraint (difference between fossil
fuel $CO_2$ emissions and the growth rate of atmospheric $CO_2$ and the uptake of $CO_2$ by oceans;
mean ± 2 standard deviation) from the 2022 Global C Budget (Friedlingstein et al., 2022). Thick
lines indicate the moving average over 10 years and thin lines indicate the annual time series. b.
Latitudinal distribution and global mean of NBP (averaged over 1980–2021) in comparison to
three datasets (CAMS (Agustí-Panareda et al., 2019), CarboScope (Rödenbeck et al., 2018), and
CT2019 (Jacobson et al., 2020)). The boxplot shows the median, interquartile range (box), and
80% percentiles (whiskers) of the global mean of NBP.

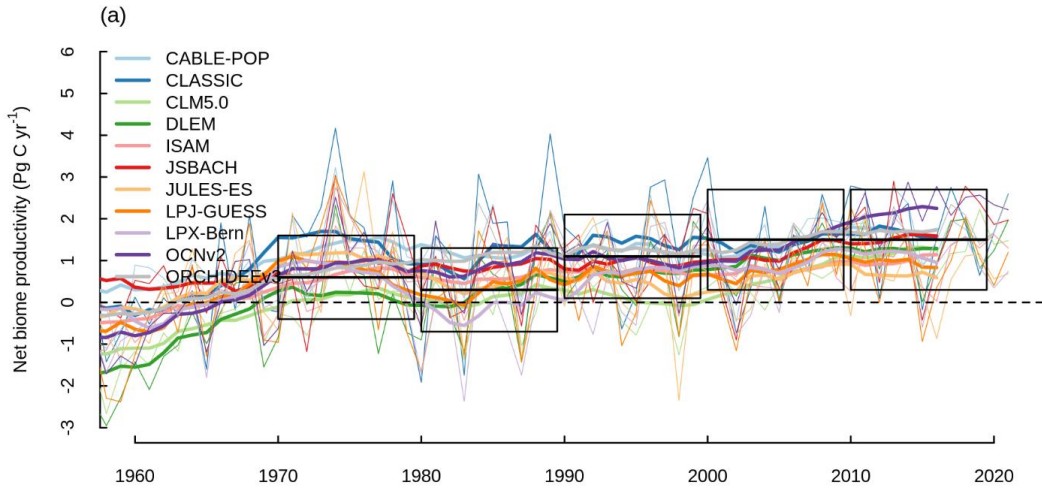

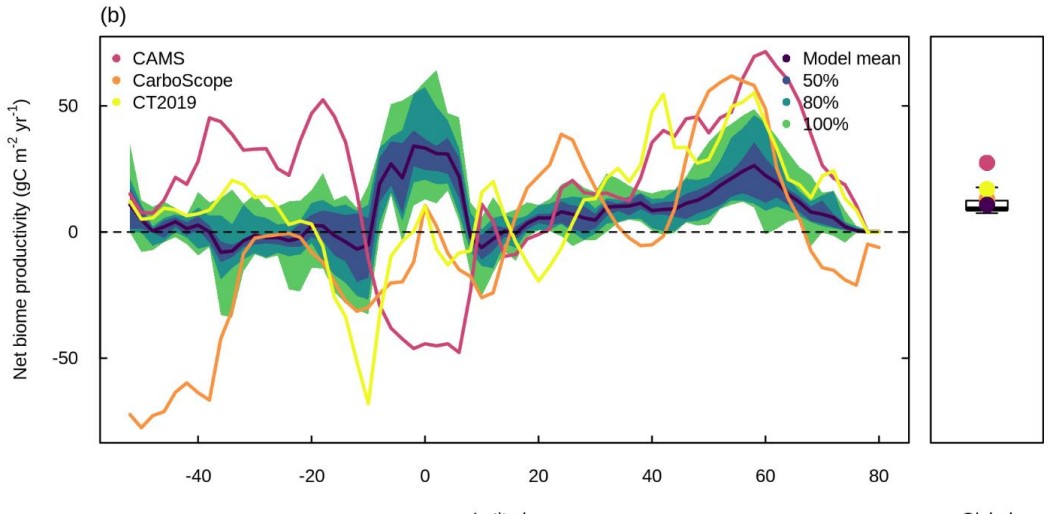




### 3.2 Overview of N cycling

Figure 3 shows a schematic of the N cycle alongside the primary N fluxes and C:N ratios of the primary pools simulated by the TRENDY-N ensemble for the present-day (averaged over 1980–2021) as well as observation-based estimates for these variables that have previously been used for model evaluation (Davies-Barnard et al., 2020). Simulated biological N fixation ranged between 20 and 566 Tg N $yr^{-1}$ (Table 2) in comparison to the observation-based estimate of 148 Tg N $yr^{-1}$, which includes both natural biological N fixation (88 Tg N $yr^{-1}$ (52 – 130 Tg N $yr^{-1}$) (Davies-Barnard and Friedlingstein, 2020)) and agricultural biological N fixation (50 – 70 Tg N $yr^{-1}$ (Herridge et al., 2008)). Simulated $N_2O$ emissions ranged between 0.9 and 11.0 Tg N $yr^{-1}$ (Table 2) in comparison to the observation-based estimate of 10.8 Tg N $yr^{-1}$ (7.1 – 16.0 Tg N $yr^{-1}$) (Tian et al., 2020). Simulated N losses (which include emissions of $NH_3$, $N_2O$, $NO_x$ and $N_2$ as well as $NO_3^-$ and $NH_4^+$ leaching) ranged between 87 and 603 Tg N $yr^{-1}$ (Table 2) in comparison to the observation-based estimate of 293 Tg N $yr^{-1}$ (Fowler et al., 2013). The simulated vegetation C:N ratio ranged between 103 and 222 (Table 2) in comparison to the observation-based estimate of 133 (Zechmeister-Boltenstern et al., 2015). The simulated combined litter-soil C:N ratio ranged between 10 and 64 (Table 2) in comparison to the observation-based estimate of 15 (Zechmeister-Boltenstern et al., 2015). Biological N fixation has the largest inter-model spread with a coefficient of variation of 1.06 (Table 2). Figure 4 shows the geographical distribution of the primary N pools and fluxes simulated by the TRENDY-N ensemble for the present-day (averaged over 1980–2021).





Figure 3: The N cycle and the primary N pools and fluxes simulated by the TRENDY-N ensemble (averaged over 1980–2021). Horizontal black lines indicate observation-based estimates that have previously been used for model evaluation (biological N fixation from Davies-Barnard and Friedlingstein (2020) and Herridge et al. (2008), vegetation and combined litter-soil C:N ratios from Zechmeister-Boltenstern et al. (2015), $N_2O$ emissions from Tian et al. (2020), and N losses from Fowler et al. (2013)). The black box indicates the terrestrial biosphere. N enters the terrestrial biosphere via biological N fixation, N deposition, and N fertilisation (entering the organic soil N pool, the inorganic soil N pool (ammonium ($NH_4^+$) or nitrate ($NO_3^-$)), or the vegetation N pool). N is transferred from the inorganic soil N pool to the vegetation N pool via N uptake. N is transferred from the vegetation N pool to the litter N pool via N litterfall. N is transferred from the litter N pool to the organic soil N pool via decomposition. N is transferred from the organic soil N pool to the inorganic soil N pool via net N mineralisation. N exits the terrestrial biosphere via N loss (which includes N leaching from soils and $N_2O$, $NO_x$, $NH_3$, and $N_2$ emissions from both soils and land use change). Not all models provide output for each N pool or flux. Note that biological N fixation simulated by LPX-Bern implicitly includes rock N sources.






Figure 4: Geographical distributions of a. vegetation N, b. litter N, c. soil N, d. biological N
fixation, e. N uptake, f. net N mineralisation, g. $N_2O$ emissions, and h. N loss simulated by the
TRENDY-N ensemble (averaged across models over 1980–2021).

(a) Vegetation N (kg N m$^{-2}$)

(b) Litter N (kg N m$^{-2}$)

(c) Soil N (kg N m$^{-2}$)

(d) Biological N fixation (g N m$^{-2}$ yr$^{-1}$)

(e) N uptake (g N m$^{-2}$ yr$^{-1}$)

(f) Net N mineralisation (g N m$^{-2}$ yr$^{-1}$)

(g) $N_2O$ emissions (g N m$^{-2}$ yr$^{-1}$)

(h) N loss (g N m$^{-2}$ yr$^{-1}$)






Table 2: Global mean and coefficient of variation of each N pool and flux simulated by the
TRENDY-N ensemble (across models over 1980–2021).

| | Coefficient of variation | Global mean | Global median | Global minimum | Global maximum |
|---|---|---|---|---|---|
| Vegetation N (Tg N) | 0.41 | 2.94 | 2.94 | 1.50 | 5.58 |
| Litter N (Tg N) | 0.81 | 1.94 | 1.08 | 0.73 | 5.61 |
| Soil N (Tg N) | 0.67 | 101.43 | 81.21 | 32.10 | 277.41 |
| Biological N fixation (Tg N yr$^{-1}$) | 1.06 | 139.63 | 101.83 | 19.92 | 565.53 |
| N uptake (Tg N yr$^{-1}$) | 0.33 | 838.78 | 698.11 | 529.53 | 1304.87 |
| Net N mineralisation (Tg N yr$^{-1}$) | 0.45 | 836.00 | 700.28 | 471.39 | 1661.53 |
| N$_2$O emissions (Tg N yr$^{-1}$) | 0.53 | 7.06 | 9.04 | 0.86 | 11.01 |
| N loss (Tg N yr$^{-1}$) | 0.85 | 187.62 | 125.96 | 87.02 | 602.77 |
| Vegetation C:N ratio | 0.28 | 159.28 | 154.50 | 102.84 | 222.22 |
| Soil C:N ratio | 0.90 | 17.32 | 11.13 | 10.00 | 63.57 |




Figure 5 shows the time series of the change from pre-industrial levels of the primary N
pools and fluxes from 1850 to 2021 simulated by the TRENDY-N ensemble. Figure 6 shows the
corresponding Kendall's tau which identifies the existence of a statistically significant trend
(Table A2). Some models suggest decreasing vegetation N (6/11 models), whereas other models
suggest increasing vegetation N (2/11 models) or no trend in vegetation N (3/11 models). Some
models suggest decreasing soil N (7/11 models), whereas other models suggest increasing soil N
(4/11 models). Some models suggest increasing biological N fixation (7/11 models), whereas
other models suggest decreasing biological N fixation (2/11 models) or no trend in biological N
fixation (2/11 models). All models suggest increasing N uptake (10/10 models). Most models
suggest increasing net N mineralisation rate (9/10 models) or no trend in N mineralisation rate
(1/10 models). All models suggest increasing $N_2O$ emissions (7/7 models) and increasing N loss
(10/10 models).





Figure 5: Time series of the change from the pre-industrial level (averaged over 1850–1870) of a.
vegetation N, b. litter N, c. soil N, d. biological N fixation, e. N uptake, f. net N mineralisation,
g. N₂O emissions, and h. N loss simulated by the TRENDY-N ensemble from 1850 to 2021.
Figure A4 shows the time series for each N pool and N flux simulated by the TRENDY-N
ensemble from 1850 to 2021.






Figure 6: Kendall's tau from the Mann-Kendall test (p-value < 0.05) for each N pool and N flux
time series simulated by the TRENDY-N ensemble from 1850 to 2021 (Table A2). A positive
value (red) indicates an increasing trend and a negative value (blue) indicates a decreasing trend
Gray indicates a statistically insignificant value and white indicates a missing value.

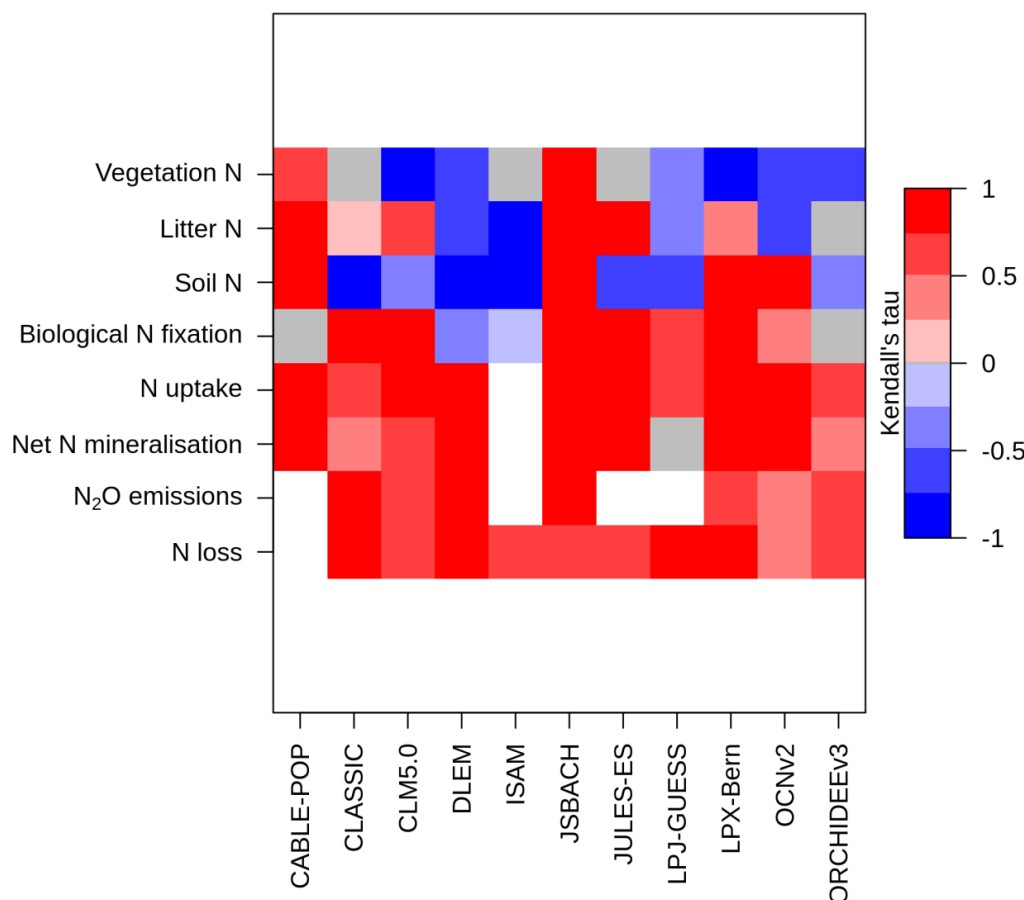




**3.3 Evaluation of biological N fixation, vegetation C:N ratio, and soil C:N ratio**

In comparison to the observation-based dataset from Davies-Barnard and Friedlingstein (2020) and the U.S. Department of Agriculture (USDA), the TRENDY-N ensemble reproduced global biological N fixation (101.8 Tg N yr$^{-1}$ vs. 108.0 Tg N yr$^{-1}$; Figure 7a and Table 2) but overestimated low-latitude biological N fixation and underestimated high-latitude biological N fixation in the Northern hemisphere (Figure 7b). In comparison to the observation-based dataset from the TRY plant trait database, the TRENDY-N ensemble overestimated the global vegetation C:N ratio (154.5 vs. 90.5; Figure 7c and Table 2) and overestimated the vegetation C:N ratio across latitudes while capturing its latitudinal pattern (Figure 7d). In comparison to the observation-based dataset from SoilGrids, the TRENDY-N ensemble overestimated the global soil C:N ratio, simulating a relatively constant soil C:N ratio across latitudes (11.1 vs. 8.8; Figure 7e and Table 2). The TRENDY-N ensemble was thus unable to capture the latitudinal pattern of the soil C:N ratio (Figure 7f).



Figure 7: Latitudinal distributions and global means of biological N fixation, vegetation C:N ratio, and soil C:N ratio simulated by the TRENDY-N ensemble (averaged across models over 1980–2021) in comparison to observations. ace. show the latitudinal distribution of the mean and boxplots show the global mean. bdf. show the latitudinal distribution of the bias. Latitudinal distributions show the mean (black line) and the 50%, 80%, and 100% percentiles across models. Boxplots show the median, interquartile range (box), and 80% percentiles (whiskers) across models. Observation-based datasets are from Davies-Barnard and Friedlingstein (2020) and the U.S. Department of Agriculture (USDA) for biological N fixation, the TRY plant trait database for vegetation C:N ratio, and SoilGrids for soil C:N ratio. LPX-Bern simulations are not shown in ab. Latitudinal distributions and global means of individual models in the TRENDY-N ensemble are shown in Figure A5.

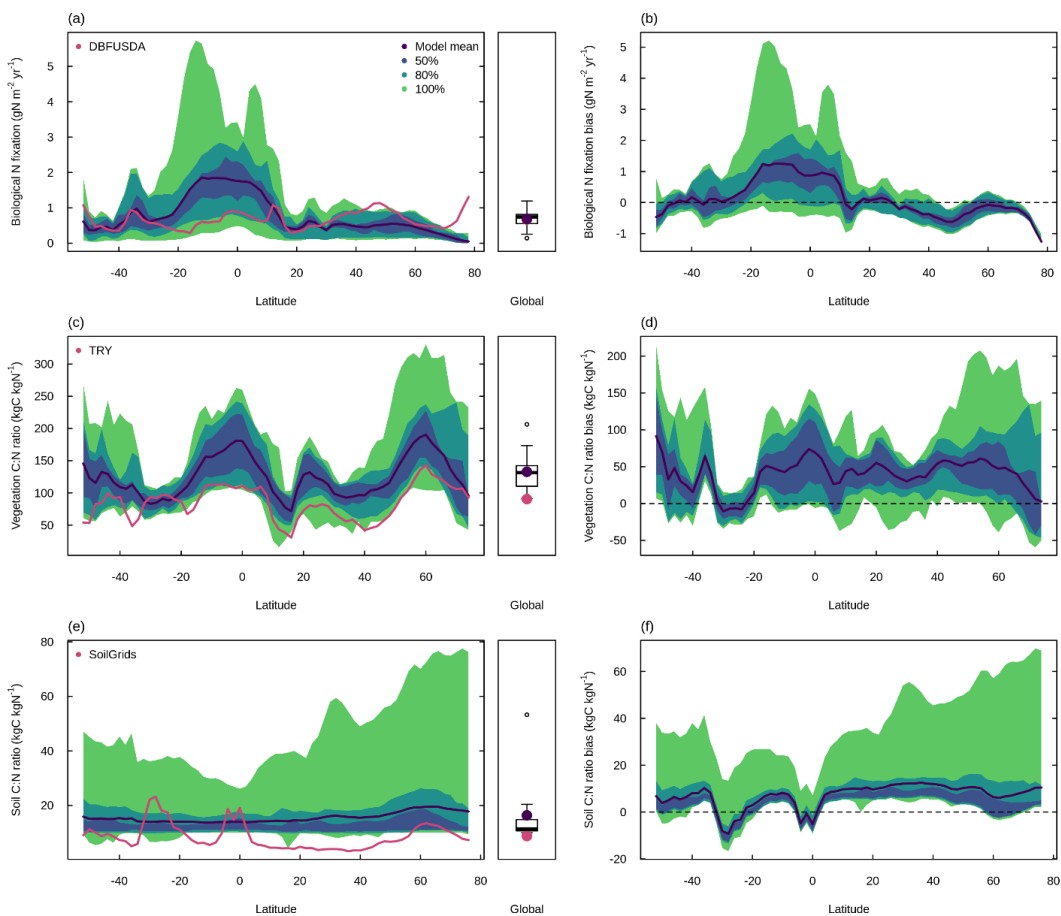



The overall score is a metric of model performance in reproducing an observation-based
dataset. Overall scores for biological N fixation, vegetation C:N ratio, and soil C:N ratio (0.46,
0.52, and 0.29 averaged across models, respectively) were lower than those for C cycling
variables (0.58 averaged across all C cycling variables and across models) (Figure 8). The mean
overall score for vegetation C:N ratio across models (0.52) was lower than the mean overall
scores for vegetation C across models (which ranged from 0.61 to 0.69 depending on the
observation-based dataset used to derive the score). Similarly, the mean overall score for soil
C:N ratio across models (0.20) was lower than the mean overall scores for soil C across models
(which ranged from 0.39 to 0.53 depending on the observation-based dataset used to derive the
score). Overall scores varied between 0.27 and 0.61 for biological N fixation, between 0.33 and
0.68 for vegetation C:N ratio, and between 0.16 and 0.39 for soil C:N ratio.



Figure 8: Overall scores of the TRENDY-N ensemble in simulating C and N cycling variables:
gross primary productivity (GPP), net biome productivity (NBP), vegetation C (CVEG), soil C
(CSOIL), leaf area index (LAI), biological N fixation (FBNF), vegetation C:N ratio (CNVEG),
and soil C:N ratio (CNSOIL). Abbreviations of the observation-based datasets are described in
the Methods and in Seiler et al. (2022).

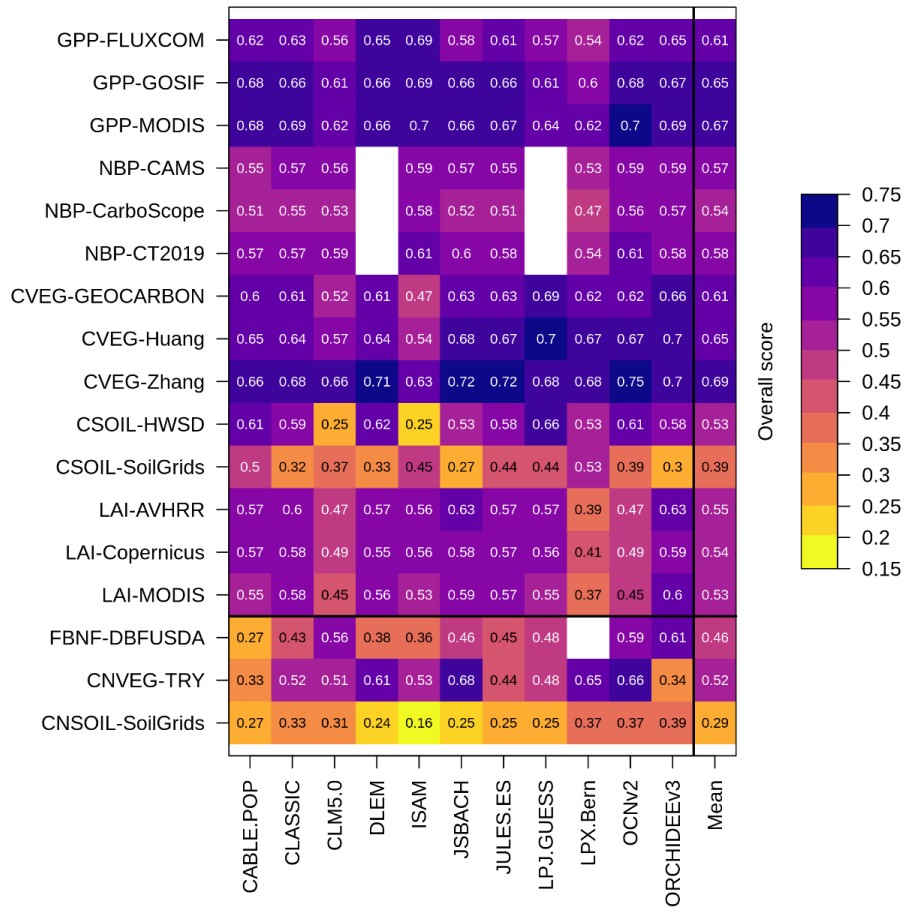






For N cycling variables, the overall score is composed of the time-mean bias score
(which assesses the difference between the time-mean of model simulations and the time-mean
of the observation-based dataset) and the spatial distribution score (which assesses the ability of
the model to reproduce the spatial pattern of the observation-based dataset) (Collier et al., 2018;
Seiler et al., 2022). For biological N fixation, the time-mean bias score averaged across models
was 0.50 and the mean spatial distribution score across models was 0.41 (Table A3). For the
vegetation C:N ratio, the time-mean bias averaged score across models was 0.47 and the mean
spatial distribution score across models was 0.58 (Table A3). For the soil C:N ratio, the time-
mean bias score averaged across models was 0.39 and the mean spatial distribution score across
models was 0.19 (Table A3).

Note that, for C fluxes, the overall score is composed of not only the time-mean bias
score and the spatial distribution score, but also the monthly centralised root-mean-square-error
score (which assesses the ability of the model to reproduce the time series of the observation-
based dataset), the seasonality score (which assess the ability of the model to reproduce the
seasonality of the observation-based dataset), and the inter-annual variability score (which
assesses the ability of the model to reproduce the inter-annual variability of the observation-
based dataset) because observation-based datasets of C fluxes are available over time (whereas
observation-based datasets of C pools and all N cycling variables are representative of the
present-day (as a single time point)).
**3.4 Representation of N cycling processes**

There were no statistically significant differences in overall scores between models with
different representations of N limitation of vegetation growth (decreasing $V_{cmax}$ and flexible C:N
stoichiometry vs. decreasing NPP), different representations of biological N fixation (function of
N limitation of vegetation growth vs. function of NPP or ET vs. time-invariant), different
representations of the response of vegetation to N limitation (dynamic vs. static), or different
representations of N limitation of decomposition (function of soil N vs. N-invariant) (Table A4).
However, models that represented decomposition as a function of soil N had a significantly
higher NBP score (corresponding to CT2019) than models that represented decomposition as N-
invariant. Similarly, there were no statistically significant differences between present-day global
values or Kendall's tau of primary C and N pools and fluxes between models with different
representations of N limitation of vegetation growth, biological N fixation, vegetation response
to N limitation, and N limitation of decomposition (Table A5 and A6). Figure A1 shows
correlations between present-day global values of the primary C and N pools and fluxes across
the TRENDY-N ensemble. Figure A2 shows correlations between Kendall's tau of the primary C
and N pools and fluxes across the TRENDY-N ensemble. Figure A3 shows correlations between
overall scores of the primary C and N pools and fluxes across the TRENDY-N ensemble.

**4 Discussion**

Despite the pivotal importance of N in constraining terrestrial C cycling and ultimately
the terrestrial C sink, there is substantial variation in simulated N cycling processes by the



terrestrial biosphere models in the TRENDY-N ensemble. The magnitude of N pools and fluxes
differ considerably between models, between 19.9 and 565.5 Tg N yr$^{-1}$ for biological N fixation
(CV = 1.1), between 1.5 and 5.6 Tg N for vegetation N (CV = 0.4), between 32.1 and 277.4 Tg
N for soil N (CV = 0.7), and between 87.0 and 602.8 Tg N yr$^{-1}$ for N loss (CV = 0.9). The spread
across the TRENDY-N ensemble suggests that approaches to represent N cycling processes vary
among terrestrial biosphere models and that there is no clear consensus yet on what the best
approaches are, supporting the use of an ensemble approach to capture the uncertainties in our
understanding of the N cycle, similarly to the C cycle (Tebaldi and Knutti, 2007).

Additionally, the historical trajectories of the N pools and fluxes differ between models:
some models simulate increasing vegetation N and soil N whereas others simulate decreasing
vegetation N and soil N between 1850 and 2021. These are the result of a host of interacting
global change drivers ($CO_2$ fertilisation, intensifying N deposition, rising temperature and
varying precipitation, land use change and associated N fertilisation regimes) whose effects are
challenging to disentangle without additional simulations. For example, while intensifying N
deposition and N fertiliser use could drive increasing soil N and N uptake, land use change could
increase N losses from both vegetation N and soil N. Despite these large differences across
models in the historical trajectories of vegetation N and soil N, all models simulate the historical
terrestrial C sink in line with observations. This suggests that the underlying N cycling processes
that regulate terrestrial C sequestration operate differently across models and may not be fully
captured. Modelled experimental manipulations (such as $CO_2$ fertilisation or N fertilisation
experiments) are imperative to evaluate model formulations of the underlying mechanisms of C-
N cycling interactions given that it is these processes that dictate the response of terrestrial C
sequestration to global change (Medlyn et al., 2015; Wieder et al., 2019; Zaehle et al., 2014).

Most models suggest increasing biological N fixation between 1850 and 2021. This
occurs either as a result of increasing vegetation biomass or the up-regulation of biological N
fixation due to N limitation imposed by $CO_2$ fertilisation or a combination thereof, depending on
the representation of biological N fixation in a given model (Table 1). This follows observations
that suggest that biological N fixation is stimulated by $CO_2$ fertilisation (Zheng et al., 2020),
although its mechanism (i.e., up-regulated biological N fixation in N-limited conditions) may not
be captured. Similarly, most models also suggest increasing N uptake between 1850 and 2021.
This also occurs as a result of increasing vegetation biomass, increasing soil N from intensifying
N deposition and N fertiliser use, or increasing biological N fixation, mycorrhizae and root
allocation due to N limitation imposed by $CO_2$ fertilisation, again dependent on the
representation of the vegetation response to N limitation in a given model (Table 1). Most
models suggest increasing net N mineralisation rate between 1850 and 2021 likely due to rising
temperature following observations (Liu et al., 2017). Most models suggest increasing $N_2O$
emissions (and N losses) between 1850 and 2021 likely due to rising temperature and
intensifying N deposition and N fertiliser use following observations (Tian et al., 2020).

We focused on three key N cycling processes for evaluation: biological N fixation,
vegetation C:N ratio, and soil C:N ratio. These three key N cycling processes have important
implications for projecting the future terrestrial C sink. Biological N fixation is the dominant





natural N supply to terrestrial ecosystems and allows vegetation to increase N uptake in N-
limited conditions, reduce N limitation, and thus sustain terrestrial C sequestration, such as in
response to N limitation imposed by $CO_2$ fertilisation (Zheng et al., 2020). Vegetation and soil
C:N ratios reflect assimilated C per unit N and thus terrestrial C sequestration. They can
potentially vary, such as in response to high photosynthesis rates relative to N uptake rates driven
by $CO_2$ fertilisation (Elser et al., 2010). Overall scores of N cycling variables, which quantify
model performance in reproducing an observation-based dataset, are lower than overall scores of
corresponding C cycling variables, suggesting that models could be less capable of capturing N
cycling processes than C cycling processes. However, this could also be due to the significant
uncertainty associated with measurements of N cycling processes as discussed below. Besides
models that represent N limitation of decomposition yielding a higher overall NBP score, there
were no statistically significant differences between models with different representations of N
limitation of vegetation growth, biological N fixation, the response of vegetation to N limitation,
and N limitation of decomposition for the overall score, present-day global value, or Kendall's
tau. This is likely due to the low number of models in the TRENDY-N ensemble and the
confounding influence of other process representations. Studies have explored the validity of
different representations of N cycling processes within a single model, suggesting that alternative
representations of a biological N fixation, ecosystem C:N stoichiometry, and ecosystem N losses
lead to substantial differences in simulated C cycling (Kou-Giesbrecht and Arora, 2022;
Meyerholt et al., 2020; Peng et al., 2020; Wieder et al., 2015a).
The TRENDY-N ensemble reproduced global observation-based biological N fixation
but tended to overestimate low-latitude biological N fixation and underestimate high-latitude
biological N fixation. This is likely because most models represented biological N fixation
phenomenologically as a function of a measure of vegetation activity (either NPP or ET). Since
there is higher vegetation activity at low latitudes than at high latitudes these models thus
represent higher biological N fixation at low latitudes than at high latitudes. However, because
biological N fixation is down-regulated in non-N-limited conditions, it is often down-regulated at
low latitudes, which are generally not (or at least less) N-limited (Barron et al., 2011; Batterman
et al., 2013; Sullivan et al., 2014). While CLASSIC, CLM5.0, and OCNv2 can represent the
down-regulation of biological N fixation in non-N-limited conditions, they still simulate high
low-latitude biological N fixation. This suggests that the strength of regulation of biological N
fixation could be insufficient and/or that there could be unaccounted N sources at low latitudes.
For example, rock N weathering could be a significant N source to terrestrial ecosystems. Some
estimates have suggested that rock N weathering could be as high as $11 - 18$ Tg N yr$^{-1}$ globally
(Houlton et al., 2018) but is not explicitly represented in the TRENDY-N ensemble (with the
exception of LPX-Bern which calculates all external N sources post hoc to simulate a closed N
cycle thereby implicitly including rock N sources). The discrepancy between modelled and
observed biological N fixation could also be due to uncertainty in the observation-based dataset
given the difficulties associated with measuring biological N fixation (Soper et al., 2021).
Ecological theory (Hedin et al., 2009) has suggested that natural biological N fixation should be
higher at low latitudes given large N losses, in contrast to the observation-based dataset from
Davies-Barnard and Friedlingstein (2020). Observational uncertainty is discussed further below.





The TRENDY-N ensemble overestimated global observation-based vegetation C:N ratio
but reproduced its latitudinal pattern (as also indicated by its higher spatial distribution score).
This is because most models represent different plant functional types (e.g., evergreen needleleaf
trees, deciduous broadleaf trees, evergreen broadleaf trees, etc.) with different tissue C:N ratios
(which can either be flexible within a constrained range or time-invariant). These plant
functional types are geographically distributed according to similar land cover products. The
TRENDY-N ensemble overestimated global observation-based soil C:N ratio and failed to
reproduce its latitudinal pattern (as also indicated by its lower spatial distribution score). In
particular, models failed to reproduce the peak at the equator and the peak at approximately -
30°S, corresponding to tropical forests and deserts respectively. This is because most models
represent a constant soil C:N ratio (both temporally and spatially) and are thus unable to capture
the spatial variability in the soil C:N ratio. Improving the representation of soil N is an important
future direction for terrestrial biosphere model development given the essential feedbacks
between soil N and soil C.
Evaluating N cycling in terrestrial biosphere models is severely restricted by the lack of
available observations of N cycling. N cycling processes are notoriously difficult to measure,
such as biological N fixation (Soper et al., 2021) and gaseous N losses (Barton et al., 2015). In
the past, N cycling has been commonly evaluated by comparison to estimates of global N pools
and fluxes derived from a small number of observations that have been scaled up or averaged to
yield a value with wide confidence intervals (Davies-Barnard et al., 2020). Not only are these
global totals highly uncertain, but they also do not allow for the analysis of spatial patterns. Here,
we present an improved framework to evaluate three key N cycling processes – biological N
fixation, vegetation C:N ratio, and soil C:N ratio – in terrestrial biosphere models. However,
these globally-gridded observation-based datasets are also uncertain, given uncertainty in the
estimates of tissue C:N ratios for different plant functional types and tissue fraction of total
biomass (especially those of roots and wood which had a lower number of measurements in
comparison to that of leaves), as well as in the measurements and models used to derive soil N
(Batjes et al., 2020). Importantly, more observations of additional N cycling processes are
necessary to fully evaluate N cycling in terrestrial biosphere models. Multiple observation-based
datasets from different sources of a given N cycling process are necessary to evaluate
observational uncertainty (Seiler et al., 2021). Observation-based datasets of N cycling processes
at intra-annual and inter-annual time scales are necessary to assess temporal patterns.
Paleoclimatic observations could also be utilised for evaluation (Joos et al., 2020). Leveraging
advances in remote sensing (Knyazikhin et al., 2013; Townsend et al., 2013) as well as
incorporating N cycling process measurements into research networks such as FLUXNET (Vicca
et al., 2018) is essential.
While some of the models in the TRENDY-N ensemble have the capability of
representing coupled C, N, and phosphorus (P) cycling (Goll et al., 2012; Nakhavali et al., 2022;
Sun et al., 2021; Wang et al., 2010, 2020b; Yang et al., 2014), P cycling was not active in the
model simulations in the GCP 2022. P limitation could be important for limiting terrestrial C
sequestration, especially in low-latitude forests (Elser et al., 2007; Terrer et al., 2019; Wieder et



al., 2015b). As more models incorporate coupled C-N-P cycling (Reed et al., 2015), observation-
based datasets of P will also be necessary for model evaluation.

**5 Conclusions**
Because the TRENDY-N ensemble overestimated both vegetation and soil C:N ratios, it
is possible that models could overestimate assimilated C per unit N and thus future terrestrial C
sequestration under $CO_2$ fertilisation. Alongside discrepancies in biological N fixation, this could
lead to biases in projections of the future terrestrial C sink by the TRENDY-N ensemble (not to
mention the other terrestrial biosphere models in the TRENDY ensemble that do not represent
coupled C-N cycling). While terrestrial biosphere models are capable of reproducing the current
terrestrial C sink, the results presented here suggest that underlying mechanisms of C-N cycling
interactions operate differently across models and may not be fully captured. These interactions
are critical for projections of the future terrestrial C sink as the C/N balance is expected to shift
in the future under interacting global change drivers.



**Code availability**

AMBER is available at https://gitlab.com/cseiler/AMBER.

**Data availability**

Biological N fixation, vegetation C:N ratio, and soil C:N ratio are available at
https://gitlab.com/siankg/amber-nitrogen.

**Author contribution**

SKG designed and conducted the study and prepared the initial manuscript. VA and CS provided
feedback on the initial manuscript and its subsequent revisions. The other co-authors conducted
TRENDY simulations and provided feedback on the manuscript.

**Competing interests**

The authors declare that they have no conflict of interest.

**Acknowledgements**

The authors would like to thank T Davies-Barnard for compiling the observations used to
evaluate biological N fixation. ORCHIDEEv3 simulations were granted access to the HPC
resources of GENCI-TGCC under the allocation A0130106328.



**Appendix A**
Table A1: IGBP land cover type, corresponding TRY plant trait database PFT, tissue C:N ratios
(from the TRY plant trait database (Kattge et al., 2020)), tissue fractions (Poorter et al., 2012),
and calculated total C:N ratio.

| IGBP land cover type | TRY plant trait database PFT | Leaf C:N | Leaf fraction | Root C:N | Root fraction | Stem C:N | Stem fraction | Total C:N |
|---|---|---|---|---|---|---|---|---|
| 0 bare | - | | | | | | | |
| 1 Evergreen needleleaf forest | Tree evergreen needleleaf<br>Temperate evergreen needleleaf<br>Boreal evergreen needleleaf<br>Gymnosperm evergreen needleleaf tree<br>Temperate conifer<br>Boreal conifer<br>Evergreen gymnosperm | 40.1 | 0.04 | 51.9 | 0.21 | 305.4 | 0.75 | 241.5 |
| 2 Evergreen broadleaf forest | Tree evergreen broadleaf<br>Temperate evergreen broadleaf<br>Tropical evergreen broadleaf<br>Boreal evergreen broadleaf<br>Angiosperm evergreen broadleaf tree<br>Gymnosperm evergreen broadleaf tree<br>Temperate evergreen<br>Rainforest<br>Evergreen angiosperm | 26.8 | 0.02 | 26.4 | 0.16 | 139.3 | 0.82 | 119.0 |
| 3 Deciduous needleleaf forest | Tree deciduous needleleaf<br>Boreal deciduous needleleaf<br>Gymnosperm deciduous needleaf tree<br>Deciduous gymnosperm | | | | | | | 241.5[a] |
| 4 Deciduous broadleaf forest | Tree deciduous broadleaf<br>Temperate deciduous broadleaf<br>Tropical deciduous broadleaf<br>Boreal deciduous broadleaf<br>Angiosperm deciduous broadleaf tree<br>Gymnosperm deciduous broadleaf tree<br>Temperate deciduous<br>Deciduous angiosperm | 21.5 | 0.03 | 39.6 | 0.21 | 102.1 | 0.76 | 86.6 |
| 5 Mixed forest | | | | | | | | 149.0[b] |



| | | | | | | | | |
|---|---|---|---|---|---|---|---|---|
| 6 Closed shrubland | Shrub evergreen broadleaf | 34.5 | 0.09 | 24.9 | 0.47 | 216.7 | 0.49 | 121.0 |
| 7 Open shrubland | Evergreen shrub Deciduous shrub Shrub | 34.5 | 0.09 | 24.9 | 0.47 | 216.7 | 0.49 | 121.0 |
| 8 Woody savannas | Angiosperm evergreen broadleaf shrub | 34.5 | 0.09 | 24.9 | 0.36 | 216.7 | 0.57 | 134.5 |
| 9 Savannas | Angiosperm deciduous broadleaf shrub Gymnosperm evergreen broadleaf shrub Desert shrub Savanna evergreen Savanna deciduous | 34.5 | 0.09 | 24.9 | 0.36 | 216.7 | 0.57 | 134.5 |
| 10 Grasslands | Grass C3 Grass C4 Temperate herbaceous Tropical herbaceous Herbaceous C3 Herbaceous C4 Angiosperm herbaceous C3 Angiosperm herbaceous C4 | 18.6 | 0.17 | 30.9 | 0.77 | 29.3 | 0.27 | 34.9 |
| 11 Permanent wetlands | | | | | | | | 34.9[c] |
| 12 Croplands | Crop C3 | 11.7 | 0.17 | 30.9[c] | 0.77 | 29.3[c] | 0.27 | 28.9 |
| 13 Urban and built-up | - | | | | | | | |
| 14 Cropland / natural vegetation mosaic | | | | | | | | 28.9[d] |
| 15 Snow and ice | - | | | | | | | |
| 16 Barren or sparsely vegetated | - | | | | | | | |

[a] Value from evergreen needleleaf forest.
[b] Average of evergreen needleleaf forest, evergreen broadleaf forest, and deciduous broadleaf forest.
[c] Value from grasslands.
[d] Value from croplands.



Table A2: Kendall's tau from the Mann-Kendall test (p-value < 0.05) for each N pool and N flux
time series simulated by the TRENDY-N ensemble from 1850 to 2021. NS indicates that
Kendall's tau is not significant. NA indicates that the variable was not reported by the model.

|  | CABLE-POP | CLASSIC | CLM5.0 | DLEM | ISAM | JSBACH | JULES-ES | LPJ-GUESS | LPX-Bern | OCNv2 | ORCHIDEEv3 |
|---|---|---|---|---|---|---|---|---|---|---|---|
| Vegetation N | 0.58 | NS | -0.97 | -0.51 | NS | 0.83 | NS | -0.25 | -0.75 | -0.67 | -0.51 |
| Litter N | 0.88 | 0.15 | 0.65 | -0.7 | -0.87 | 0.92 | 0.86 | -0.35 | 0.44 | -0.69 | NS |
| Soil N | 1 | -0.8 | -0.47 | -0.97 | -0.91 | 0.99 | -0.67 | -0.68 | 1 | 1 | -0.3 |
| Biological N fixation | NS | 0.95 | 0.84 | -0.33 | -0.11 | 0.89 | 0.79 | 0.62 | 0.92 | 0.45 | NS |
| N uptake | 0.89 | 0.64 | 0.81 | 0.78 | NA | 0.81 | 0.85 | 0.54 | 0.82 | 0.85 | 0.71 |
| Net N mineralisation | 0.91 | 0.33 | 0.73 | 0.87 | NA | 0.85 | 0.76 | NS | 0.86 | 0.82 | 0.31 |
| N$_2$O emissions | NA | 0.92 | 0.7 | 0.87 | NA | 0.95 | NA | NA | 0.7 | 0.42 | 0.69 |
| N loss | NA | 0.94 | 0.67 | 0.94 | 0.73 | 0.59 | 0.63 | 0.94 | 0.81 | 0.42 | 0.65 |




Table A3: Time-mean bias score ($S_{bias}$), spatial distribution score ($S_{dist}$), and overall score
($S_{overall}$) of the TRENDY-N ensemble in simulating biological N fixation, vegetation C:N ratio,
and soil C:N ratio.

| | Biological N fixation | | | Vegetation C:N ratio | | | Soil C:N ratio | | |
|---|---|---|---|---|---|---|---|---|---|
| | $S_{bias}$ | $S_{dist}$ | $S_{overall}$ | $S_{bias}$ | $S_{dist}$ | $S_{overall}$ | $S_{bias}$ | $S_{dist}$ | $S_{overall}$ |
| CABLE-POP | 0.46 | 0.08 | 0.27 | 0.34 | 0.33 | 0.33 | 0.2 | 0.34 | 0.27 |
| CLASSIC | 0.46 | 0.4 | 0.43 | 0.45 | 0.59 | 0.52 | 0.43 | 0.22 | 0.33 |
| CLM5.0 | 0.55 | 0.56 | 0.56 | 0.57 | 0.46 | 0.51 | 0.45 | 0.16 | 0.31 |
| DLEM | 0.46 | 0.29 | 0.38 | 0.47 | 0.75 | 0.61 | 0.48 | 0.01 | 0.24 |
| ISAM | 0.47 | 0.24 | 0.36 | 0.49 | 0.57 | 0.53 | 0.05 | 0.28 | 0.16 |
| JSBACH | 0.48 | 0.44 | 0.46 | 0.63 | 0.74 | 0.68 | 0.38 | 0.11 | 0.25 |
| JULES-ES | 0.47 | 0.43 | 0.45 | 0.4 | 0.49 | 0.44 | 0.51 | 0 | 0.25 |
| LPJ-GUESS | 0.51 | 0.45 | 0.48 | 0.45 | 0.52 | 0.48 | 0.49 | 0.01 | 0.25 |
| LPX-Bern | NA | NA | NA | 0.54 | 0.76 | 0.65 | 0.33 | 0.4 | 0.37 |
| OCNv2 | 0.56 | 0.62 | 0.59 | 0.56 | 0.76 | 0.66 | 0.47 | 0.26 | 0.37 |
| ORCHIDEEv3 | 0.6 | 0.63 | 0.61 | 0.27 | 0.41 | 0.34 | 0.48 | 0.31 | 0.39 |




Table A4: Overall scores of biological N fixation, vegetation C:N ratio, soil C:N ratio, and NBP
of TRENDY-N ensemble models with different representations of key N cycling processes (N
limitation of vegetation growth, biological N fixation, vegetation response to N limitation, and N
limitation of decomposition, see Table 1).

| | | BNF-DBFUSDA | CNVEG-TRY | CNSOIL-SoilGrids | NBP-CAMS | NBP-Carboscope | NBP-CT2019 |
|---|---|---|---|---|---|---|---|
| N limitation of vegetation growth | $V_{cmax}$ / flexible C:N stoichiometry | 0.49 | 0.47 | 0.32 | 0.57 | 0.54 | 0.58 |
| | NPP | 0.41 | 0.58 | 0.26 | 0.56 | 0.52 | 0.58 |
| | p-value | 0.21 | 0.14 | 0.15 | 0.59 | 0.44 | 0.9 |
| Biological N fixation | f(N limitation of vegetation growth) | 0.44 | 0.34 | 0.33 | 0.57 | 0.54 | 0.57 |
| | f(NPP) or f(ET) | 0.44 | 0.53 | 0.23 | 0.57 | 0.54 | 0.6 |
| | Time-invariant | 0.53 | 0.56 | 0.33 | 0.57 | 0.55 | 0.59 |
| | p-value | 0.59 | 0.07 | 0.06 | 0.92 | 0.91 | 0.28 |
| Vegetation response to N limitation | Dynamic | 0.49 | 0.56 | 0.3 | 0.57 | 0.55 | 0.59 |
| | Static | 0.43 | 0.5 | 0.28 | 0.56 | 0.53 | 0.58 |
| | p-value | 0.44 | 0.41 | 0.71 | 0.48 | 0.3 | 0.67 |
| N limitation of decomposition | f(soil N) | 0.47 | 0.57 | 0.26 | 0.57 | 0.54 | 0.6 |
| | N-invariant | 0.45 | 0.46 | 0.32 | 0.56 | 0.52 | 0.56 |
| | p-value | 0.86 | 0.17 | 0.16 | 0.26 | 0.44 | 0.02 |






Table A5: Present-day global values of biological N fixation, vegetation C:N ratio, and soil C:N
ratio simulated by TRENDY-N ensemble models with different representations of key N cycling
processes (N limitation of vegetation growth, biological N fixation, vegetation response to N
limitation, and N limitation of decomposition, see Table 1).

|  |  | Biological N fixation | Vegetation C:N ratio | Soil C:N ratio |
|---|---|---|---|---|
| N limitation of vegetation growth | $V_{cmax}$ / flexible C:N stoichiometry | 106.78 | 161.8 | 12.75 |
|  | NPP | 179.06 | 156.26 | 22.79 |
|  | p-value | 0.51 | 0.85 | 0.39 |
| Biological N fixation | f(N limitation of vegetation growth) | 123.14 | 201.68 | 15.71 |
|  | f(NPP) or f(ET) | 66.37 | 177.37 | 24.31 |
|  | Time-invariant | 118.95 | 123.89 | 11.64 |
|  | p-value | 0.27 | 0.15 | 0.68 |
| Vegetation response to N limitation | Dynamic | 99.25 | 143.32 | 11.22 |
|  | Static | 173.29 | 172.58 | 22.4 |
|  | p-value | 0.41 | 0.29 | 0.24 |
| N limitation of decomposition | f(soil N) | 88.21 | 153.36 | 20.04 |
|  | N-invariant | 201.34 | 166.38 | 14.04 |
|  | p-value | 0.3 | 0.66 | 0.53 |






Table A6: Kendall's tau from the Mann-Kendall test (p-value < 0.05) for biological N fixation,
vegetation C:N ratio, and soil C:N ratio simulated by TRENDY-N ensemble models with
different representations of key N cycling processes (N limitation of vegetation growth,
biological N fixation, vegetation response to N limitation, and N limitation of decomposition, see
Table 1).

| | | Biological N fixation | Vegetation C:N ratio | Soil C:N ratio |
|---|---|---|---|---|
| N limitation of vegetation growth | $V_{cmax}$ / flexible C:N stoichiometry | 0.48 | -0.01 | -0.04 |
| | NPP | 0.43 | -0.74 | 0 |
| | p-value | 0.89 | 0.06 | 0.94 |
| Biological N fixation | f(N limitation of vegetation growth) | 0 | -0.31 | 0.02 |
| | f(NPP) or f(ET) | 0.55 | -0.6 | 0.14 |
| | Time-invariant | 0.74 | 0.39 | -0.03 |
| | p-value | 0.15 | 0.15 | 0.97 |
| Vegetation response to N limitation | Dynamic | 0.5 | -0.08 | 0.01 |
| | Static | 0.41 | -0.56 | -0.04 |
| | p-value | 0.77 | 0.3 | 0.93 |
| N limitation of decomposition | f(soil N) | 0.42 | -0.42 | 0.31 |
| | N-invariant | 0.5 | -0.25 | -0.42 |
| | p-value | 0.8 | 0.7 | 0.14 |






Figure A1: Correlations between present-day global values (averaged over 1980–2021) of primary C and N pools and fluxes across TRENDY-N ensemble models: vegetation C (CVEG), litter C (CLITTER), soil C (CSOIL) ), net biome productivity (NBP), gross primary productivity (GPP), autotrophic respiration (RA), heterotrophic respiration (RH), leaf area index (LAI), vegetation N (NVEG), litter N (NLITTER), soil N (NSOIL), biological N fixation (FBNF), N uptake (NUP), net N mineralisation (NETNMIN), $N_2O$ emissions (N2O), N loss (NLOSS), vegetation C:N ratio (CNVEG), and soil C:N ratio (CNSOIL). Spearman's rank correlation coefficient is shown for statistically significant correlations (p-value < 0.05).

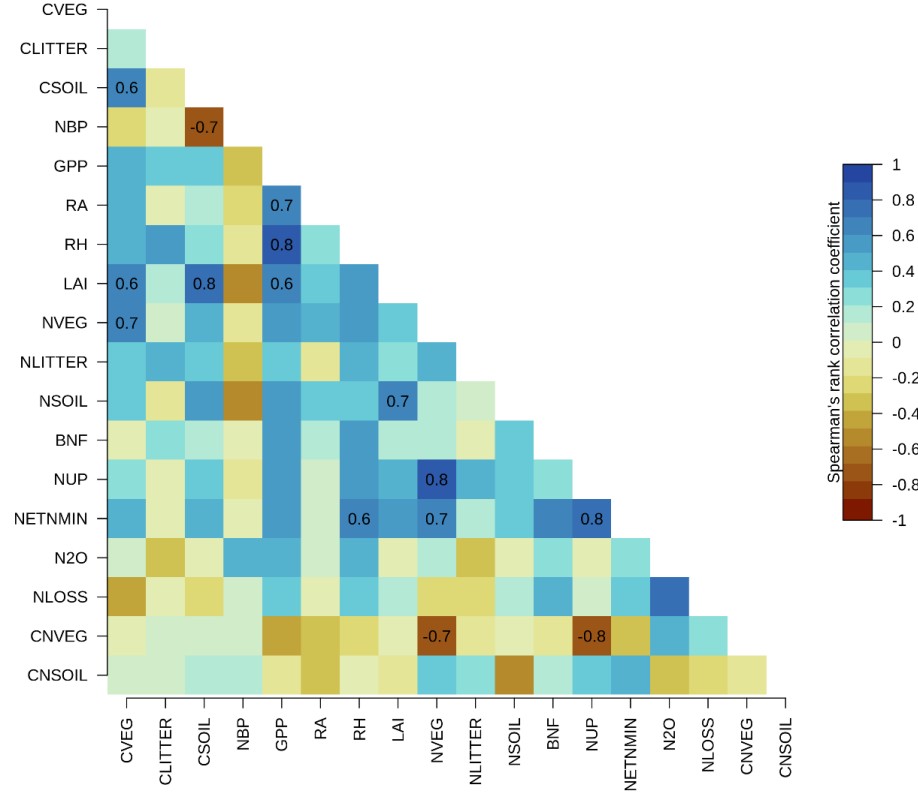

660



Figure A2: Correlations between Kendall's tau of primary C and N pools and fluxes across
TRENDY-N ensemble models: vegetation C (CVEG), litter C (CLITTER), soil C (CSOIL), net
biome productivity (NBP), gross primary productivity (GPP), autotrophic respiration (RA),
heterotrophic respiration (RH), leaf area index (LAI), vegetation N (NVEG), litter N
(NLITTER), soil N (NSOIL), biological N fixation (FBNF), N uptake (NUP), net N
mineralisation (NETNMIN), $N_2O$ emissions (N2O), N loss (NLOSS), vegetation C:N ratio
(CNVEG), and soil C:N ratio (CNSOIL). Spearman's rank correlation coefficient is shown for
statistically significant correlations (p-value < 0.05).

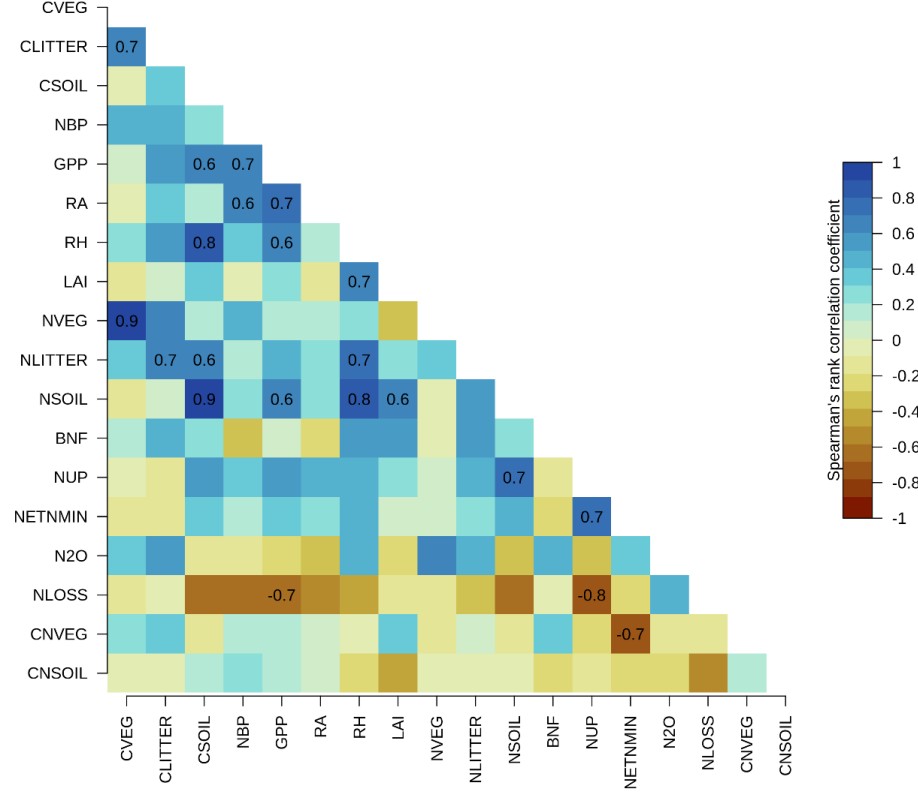



Figure A3: Correlations between overall scores of primary C and N pools and fluxes across
TRENDY-N ensemble models: gross primary productivity (GPP), net biome productivity (NBP),
vegetation C (CVEG), soil C (CSOIL), leaf area index (LAI), biological N fixation (FBNF),
vegetation C:N ratio (CNVEG), and soil C:N ratio (CNSOIL). Abbreviations of the observation-
based datasets are described in the Methods and in (Seiler et al., 2022). Spearman's rank
correlation coefficient is shown for statistically significant correlations (p-value < 0.05).

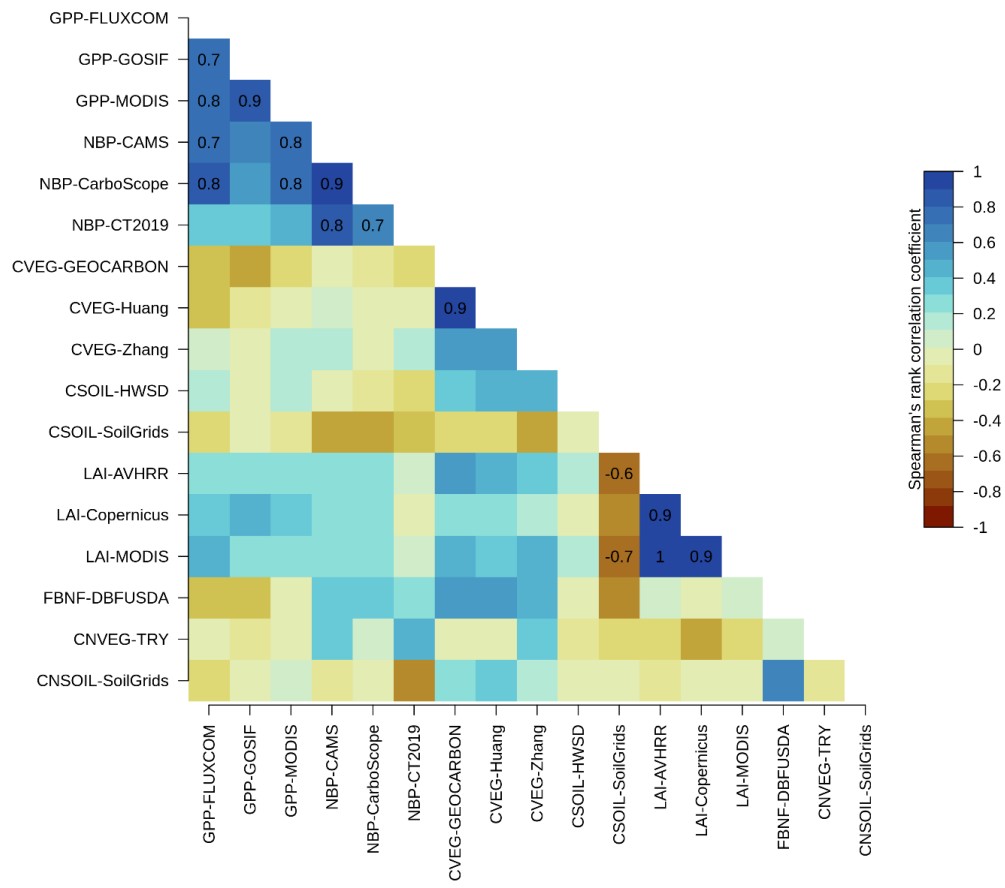






Figure A4: Time series of a. vegetation N, b. litter N, c. soil N, d. biological N fixation, e. N
uptake, f. net N mineralisation, g. N$_2$O emissions, and h. N loss simulated by the TRENDY-N
ensemble from 1850 to 2021.






Figure A5: Latitudinal distributions and global means of ab. biological N fixation, cd. vegetation
C:N ratio, and ef. soil C:N ratio simulated by the TRENDY-N ensemble (averaged across models
over 1980–2021) in comparison to observation-based datasets from (Davies-Barnard and
Friedlingstein, 2020) for biological N fixation, the TRY plant trait database for vegetation C:N
ratio, and SoilGrids for soil C:N ratio. Boxplots show the median, interquartile range (box), and
80% percentiles (whiskers) of the global mean.

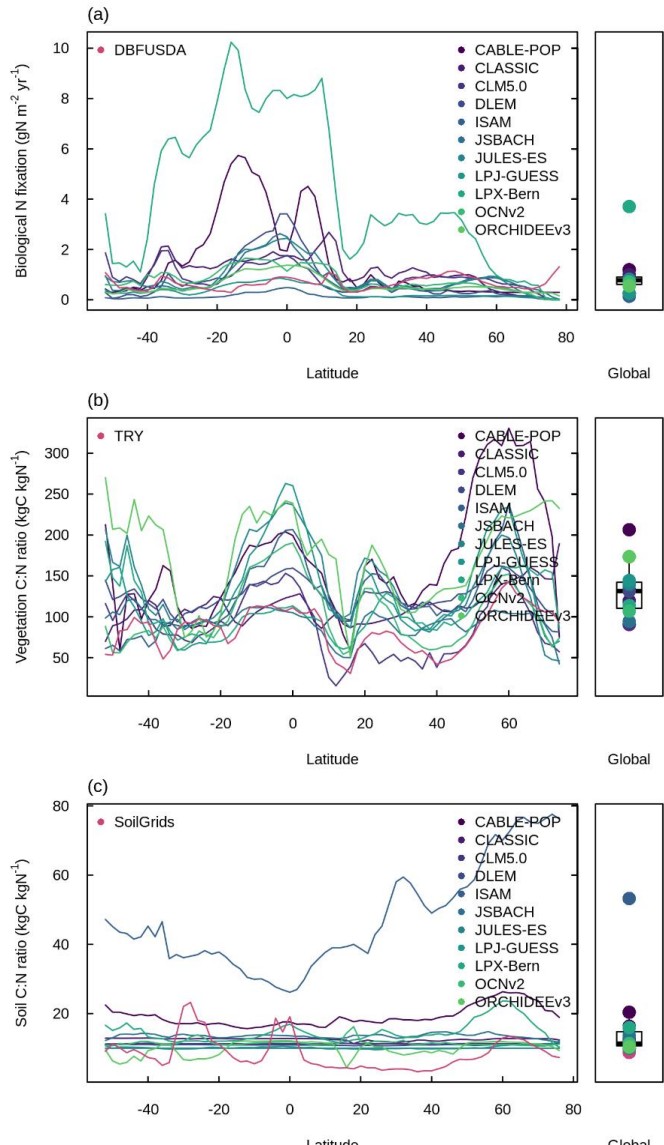




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
