# Peer review of "Evaluating Nitrogen Cycling in Terrestrial Biosphere Models: a Disconnect between the"

_EGUsphere, 2023_

## Author Response (AR1)

Dear authors,

Thank you for submitting your response. I was surprised to learn from the editorial system that the peer-review was stopped because of lack of response from you. I am glad that the manuscript is back in play. The manuscript is potentially an impactful paper on an interesting topic. But it needs major revisions for further considerations. Your finding that the C and N dynamics in TEMs could be disconnected is very interesting. It would be great if you could dig deeper and try to identify the cause of the problem and suggest potential solutions. Also, the title could be improved to better reflect your findings. I now invite you to submit a revised manuscript addressing the issues raised by the peer-review.

Sincerely

Somnath
* * *
**Dear Dr. Somnath Baidya Roy,**

**Thank you to you and the reviewers for your insightful comments. We appreciate that the reviewers were generally favourable towards the manuscript. Their comments were extremely helpful in improving our paper. Below we provide point-by-point responses to these comments.**

**In particular, we have emphasized the disconnect between C and N cycling in the TRENDY models and we have revised the title of the manuscript accordingly. We have added additional analysis of this disconnect to the main text of the manuscript.**

**Thank you for your consideration and we look forward to hearing from you.**

**Sincerely,**

**Dr. Sian Kou-Giesbrecht (on behalf of all authors)**

Reviewer comment 1

This is a well-written assessment of modeled nitrogen cycle outputs from the latest TRENDY. Overall, it's a good reference in documenting these outputs and the figures are good. The paper reads more like documentation than process-level science advance, but I think that's fine for EGUsphere. As often with these analyses, the outcomes are that there is model spread and uncertainty, but the mechanisms and implications are somewhat opaque. Although, the science advance could be gleaned from the alarming statements (see below) scattered throughout as a bit of a call to arms. Perhaps change the paper title to be more impactful / less vague after the colon to reflect some of these alarming findings.

The striking statements to me included those that indicated no difference in C sink with and without N cycling; no score differences among models with different representations of BNF, N limitation to growth, decomposition, etc.; and, models generally reproduced the historical C sink despite huge variability in N pools/fluxes, and other seemingly glaring issues like constant soil C:N.

These statements are alarming and disconnect from the authors' statements that N cycling should be important; but, the statements made by the authors above suggest otherwise—N cycling is unimportant, as the models will do whatever they do seemingly disconnected from N cycling (e.g., they're tuned to the C sink).

**AUTHOR RESPONSE: We have rewritten the manuscript to emphasize that a main result of our analysis is the significant spread between the models in simulating N cycling processes despite their ability to all reproduce the historical terrestrial C sink and that this suggests a disconnect between the C and N cycles. We have changed the title of the manuscript and added a new section to the Discussion called "Disconnect between C and N cycling in terrestrial biosphere models" to specifically dive into this result (Lines 546 to 571). As you suggested, we first explain that N cycling should be important (given substantial empirical evidence) but that our results suggest a disconnect between the C and N cycles in models because they are calibrated against the C cycle. We then explain that this disconnect is critical for future analysis and model development in another new section of the Discussion called "Future directions" (Lines 572 to 615). In particular, we explain that, while this disconnect is not apparent in historical simulations, it will become particularly consequential for projecting the terrestrial C sink under future global change which is likely to modify the C-N balance through N limitation of CO2 fertilisation and intensifying N deposition among other drivers of global change. By dedicating a section of the Discussion to clearly explain the disconnect between the C and N cycles we hope that we have clarified our results. We have also clarified our principal results in the Abstract, Introduction and Conclusions.**

So, this makes me wonder 2 things: 1) is the N cycle really just totally decoupled from the C cycle in the models; and/or, 2) are the authors performing the right tests to understand the sensitivity of the C cycle from the N cycle. For #2, the authors focus on BNF, and Veg and Soil

C:N. It seems that the tests need to go another step further in some sort of normalization into NBP. The tests as presented seem indirect, but then the authors make bold statements about the importance, making it hard to trace the justification. Were there tests on progressive N limitation? Tipping points? Issues with N fertilization? Etc. If there is constant soil C:N, shouldn't this manifest somewhere problematic in the C sink?

**AUTHOR RESPONSE: We have added and highlighted two additional analyses in the main text (described in Lines 441 to 462). First, we examine correlations between scores of model performance in simulating C cycling and scores of model performance in simulating N cycling (Figure A2). These scores include both a time-mean bias component and a spatial distribution component (described in Lines 265 to 278). Unfortunately, because the observation-based datasets for biological N fixation, vegetation C:N, and soil C:N are representative of the present-day and do not have a time dimension, we cannot evaluate seasonality or inter-annual variability in comparison to that of net biome productivity. Second, we examine significant differences in model performance between models with different representations of fundamental N cycling processes (N limitation of vegetation growth, biological N fixation, vegetation response to N limitation, and N limitation of decomposition) (Table A4). Unfortunately, neither of these analyses yielded significant results. As you pointed out, this is often the case with model intercomparisons and is likely due to the low number of models and the confounding influence of other process representations (Lines 461 to 462). Alongside the significant spread between simulated N cycling processes, we interpret these results as pointing to the disconnect between C and N cycling (described above). In the new version of the manuscript, we have emphasized this as our main result rather than explicit implications for the future terrestrial C sink. Additionally, we have added a new "Future directions" section to the Discussion that explains that the best test of nutrient limitation is modelled experimental manipulations, such as CO2 fertilisation and N fertilisation experiments (Lines 599 to 608). Here, we suggest that "a robust test of the simulated response to $CO_2$ fertilisation and N fertilisation across models would be ideal for evaluating the ability of models to represent the regulation of C cycling by N cycling under global change and thus their ability to realistically simulate the future terrestrial C sink".**

Minor comments:

L180. CLM5.0 increases BNG with N limitation only if there's enough C to pay for it [Fisher et al., 2010; Shi et al., 2016; Fisher et al., 2019].

**AUTHOR RESPONSE: We have updated the description of CLM5.0 (Lines 184-186).**

L578. Could be that new hyperspectral remote sensing could provide a nice constraint on Canopy N (e.g., SBG) [Cawse-Nicholson et al., 2021]. See also product from Fisher et al. [2012].

**AUTHOR RESPONSE: We have updated our analysis using the remote sensing leaf N content product from Moreno-Martinez et al. (2018) and used the TRY data to scale from leaf N**

**content to vegetation N content with PFT-specific relationships. Our results did not change substantially.**

L581. See also Braghiere et al. [2022].

**AUTHOR RESPONSE: We have added a reference to ELM.**

Good work overall!

**AUTHOR RESPONSE: Thank you!**

Josh Fisher

Reviewer comment 2

The paper evaluates 11 TRENDY models for simulating nitrogen cycle processes and compares them in detail with different available datasets. The paper is very well structured, well written with a relevant and topical theme.  The study does a great job in comparing the different models and the analysis is comprehensive and statistically sound.

**AUTHOR RESPONSE: Thank you!**

However, the paper fails to provide an overall understanding of the importance of nitrogen cycle in estimating carbon fluxes given the estimates it gives. Since the importance of nitrogen in the carbon cycle processes has been highlighted in the text in multiple places, statements that contradict this are also a part of the text. This leaves the reader puzzled with two main questions:

If the different TRENDY models (with no or some/different N cycle processes built in) can estimate C fluxes within a certain (acceptable) range despite estimating N fluxes that have a wide range of magnitude, what is the significance of integrating N cycle processes in the models? This has not been answered or proved anywhere in the text and is one of the major drawbacks of the paper.

**AUTHOR RESPONSE: We have added a new section to the Discussion called "Disconnect between C and N cycling in terrestrial biosphere models" that clarifies our results (Lines 546 to 571). As described in our response to Reviewer 1 above, we first explain that N cycling should be important and that this is supported by substantial empirical evidence. Then we explain that, because the models all reproduce the historical terrestrial C sink yet exhibit a substantial spread in simulated N cycling processes, our results suggest a disconnect between the C and N cycles in models. This is likely because models are calibrated to C cycling. We emphasize that this disconnect between the C and N cycles in models is the main result of our paper and that it is critical for future analysis and model development. This is described in another new section of the Discussion called "Future directions" (Lines 572 to 615), in response to your comment below. We explain that, while this disconnect between the C and N cycles is not apparent in historical simulations, it will become particularly consequential for projecting the terrestrial C sink under future global change which is likely to modify the C-N balance through N limitation of CO2 fertilisation and intensifying N deposition among other drivers of global change. We have also clarified this result in the Abstract, Introduction and Conclusions.**

Different models have implemented N cycle processes in different ways. The study could have made a prominent impact if it was able to identify which specific N cycle processes are crucial to implement in the models so that the N fluxes and pool estimates are comprehensively represented in the models and the resulting N pools and fluxes are better correlated to the observed datasets. This was probably a low hanging fruit for this study and could have been a significant contribution of the paper.

**AUTHOR RESPONSE: We have added and highlighted two additional analyses in the main text (described in Lines 441 to 462). First, we examine correlations between scores of model performance in simulating C cycling and scores of model performance in simulating N cycling (Figure A2). Second, we examine significant differences in model performance between models with different representations of fundamental N cycling processes (N limitation of vegetation growth, biological N fixation, vegetation response to N limitation, and N limitation of decomposition) (Table A4). Unfortunately, neither of these analyses yielded significant results. This is likely due to the low number of models and the confounding influence of other process representations (Lines 461 to 462). However, we point to studies that have explored the validity of different representations of N cycling processes within a single model. These studies suggest that alternative representations of a biological N fixation, ecosystem C:N stoichiometry, and ecosystem N losses lead to substantial differences in simulated C cycling (Lines 564 to 568). Alongside the significant spread between simulated N cycling processes, we interpret this lack of significance between different models as pointing to the disconnect between C and N cycling. We discuss this in Lines 560 to 564.**

Some minor points:

Fig 2b.): NBP estimates from CarboScope and CAMS vary a lot from other datasets/observations in different range of latitudes. Please add an explanation in the text for this difference in observed datasets.

**AUTHOR RESPONSE: We have added an explanation in Lines 301 to Lines 304.**

Fig 4: The plots represent average values of different N pools from the models. It would be helpful to add another set of similar plots with uncertainty ranges for estimates so that we can identify if there are specific regions where the estimates from different models are in a similar range and regions where the models produce very different numbers.

**AUTHOR RESPONSE: We have added a figure that shows variation across models (Figure A1). This figure helps explain the spread across models in simulating N cycling processes.**

The Discussion and Conclusions sections should be edited to add the potential directions of future research, given the extensive analysis shown in the paper, in a less vague manner.

**AUTHOR RESPONSE: We have added a new section to the Discussion called "Future directions" (Lines 572 to 615) that clearly suggests needed avenues for future research.**

Overall, it's a good publication that summarizes the current state of the nitrogen cycle in state-of-the-art TRENDY models. Good effort by the authors!

**AUTHOR RESPONSE: Thank you!**